# Tuning interdomain conjugation to enable *in situ* population modification in yeasts

Kevin R. Stindt,[1,2] Megan N. McClean[1,3]

**ABSTRACT**  The ability to modify and control natural and engineered microbiomes is essential for biotechnology and biomedicine. Fungi are critical members of most microbiomes, yet technology for modifying the fungal members of a microbiome has lagged far behind that for bacteria. Interdomain conjugation (IDC) is a promising approach, as DNA transfer from bacterial cells to yeast enables *in situ* modification. While such genetic transfers have been known to naturally occur in a wide range of eukaryotes and are thought to contribute to their evolution, IDC has been understudied as a technique to control fungal or fungal-bacterial consortia. One major obstacle to the widespread use of IDC is its limited efficiency. In this work, we manipulated metabolic and physical interactions between genetically tractable *Escherichia coli* and *Saccharomyces cerevisiae* to control the incidence of IDC. We test the landscape of population interactions between the bacterial donors and yeast recipients to find that bacterial commensalism leads to maximized IDC, both in culture and in mixed colonies. We demonstrate the capacity of cell-to-cell binding via mannoproteins to assist both IDC incidence and bacterial commensalism in culture and model how these tunable controls can predictably yield a range of IDC outcomes. Furthermore, we demonstrate that these controls can be utilized to irreversibly alter a recipient yeast population, by both "rescuing" a poor-growing recipient population and collapsing a stable population via a novel IDC-mediated CRISPR/Cas9 system.

**IMPORTANCE**  Fungi are important but often unaddressed members of most natural and synthetic microbial communities. This work highlights opportunities for modifying yeast microbiome populations through bacterial conjugation. While conjugation has been recognized for its capacity to deliver engineerable DNA to a range of cells, its dependence on cell contact has limited its efficiency. Here, we find "knobs" to control DNA transfer, by engineering the metabolic dependence between bacterial donors and yeast recipients and by changing their ability to physically adhere to each other. Importantly, we functionally validate these "knobs" by irreversibly altering yeast populations. We use these controls to "rescue" a failing yeast population, demonstrate the capacity of conjugated CRISPR/Cas9 to depress or collapse populations, and show that conjugation can be easily interrupted by disrupting cell-to-cell binding. These results offer building blocks toward *in situ* mycobiome editing, with significant implications for clinical treatments of fungal pathogens and other fungal system engineering.

**KEYWORDS**  conjugation, fungi, synthetic consortia, CRISPR, crossfeeding, expansion assay, population modeling, mycobiome, horizontal gene transfer, microbiome

Extraordinary advances have been made in recent years elucidating the composition and function of microbiome members, but the vast majority of this work has focused on bacterial species and often overlooks fungal participants (1). Though the number of fungal cells is typically dwarfed by bacterial cells, fungi play important roles in both

Address correspondence to Megan N. McClean, mmcclean@wisc.edu.

The authors declare no conflict of interest.

See the funding table on p. 18.

human (2–6) and environmental (7–10) microbiomes. Many fungal pathogens live as commensals in humans before becoming infectious, whether due to hospital-derived nosocomial infections or auto-immune disorders, both of which are on the rise (5). *Candida* species, which usually exist as commensals in the gut microbiome, cause many such nosocomial infections, resulting in a range of candidiasis symptoms that can lead to sepsis (2). Common skin microbiome residents in the *Malassezia* genus (6) have been implicated in Crohn's disease (3) and tumorigenesis (4). Moreover, many fungi infect plant (7) or other animal species (8), e.g., bats (9) and amphibians (10), often causing significant agricultural or environmental loss. In addition to their roles in the natural environment, the unique metabolic capabilities of fungi make them important members in food production (11, 12) and engineered bioproduction (13, 14) and bioremediation (15), often with other species in consortia, emulating the division of labor found in microbiomes. Thus, fungal microbes play important roles in human, plant, and environmental microbiomes, in addition to synthetic microbial consortia, and therefore tools for modifying the fungal microbiome are critical for advances in all these areas.

Modification of microbiome members is most frequently done by methods designed to reduce or kill off specific populations in a targeted or semi-targeted way (16). Pre- and probiotics, antibiotics and antifungals, microbial transplants, and phages seek to promote or eliminate specific populations. In addition to focusing primarily on eliminating specific populations, few of these tools are able to modify fungal microbiome members. In contrast, bacterial conjugation, a naturally occurring form of horizontal gene transfer (HGT) (17), allows genetic modification of bacterial populations instead of simple killing and has already been used for probiotics (18), defense against antibiotic-resistant pathogens (19), crop modification for desired traits (20), control of undomesticated microbial species (21), circuit-like control of synthetic consortia (22), and *in situ* microbiome engineering (23, 24). Bacterial HGT has also been shown to protect functional stability of diverse microbial communities, mitigating threats posed by compositional variations (25). Bacteria also conjugate with a variety of eukaryotic recipient cells, most commonly from bacterial donor *Agrobacterium tumefaciens* to plant cells (26). And while *A. tumefaciens* is uniquely well studied for performing interdomain conjugation (IDC) in the wild, highly genetically tractable bacteria such as *Escherichia coli* can be modified to perform IDC with diatoms (27–30), mammalian cells (31), and multiple yeast species (32–36) offering a powerful opportunity for modifying yeast *in situ*. IDC has typically been referred to as transkingdom conjugation, though this nomenclature predates (32) the domain designation of prokaryotes and eukaryotes proposed by Woese (37), which further highlights the significance of this genetic transfer mechanism between cells of different domains.

Conjugative transfer of DNA occurs in multiple stages in the bacterial cell. First, a complex of proteins called the "relaxosome," containing catalytic relaxases, nicks the conjugative plasmid at the origin of transfer ($ori^T$) and transfers one strand of the plasmid DNA to the membrane-bound type IV secretion system (T4SS) (38). The T4SS transports the relaxosome-DNA complex through both bacterial membranes and a pilus connecting the donor and recipient cells. For *E. coli* T4SS, the DNA re-circularizes in the recipient cell to recreate the original plasmid (39). Conjugation can occur via either a *cis* mechanism, in which the plasmid carrying the relaxosome genes itself contains an $ori^T$ and thus is transferred to a recipient cell or a *trans* mechanism, in which the $ori^T$ is on a separate plasmid, which gets transferred (40) (Fig. 1a).

IDC is currently limited as a tool for mycobiome modification by its relatively low efficiency (41). In fact, the vast majority of conjugation research has focused on lowering efficiency further (41, 42), in an effort to prevent the spread of antibiotic resistance, which occurs through conjugative transfer of resistance-coding genes (43). And while it is often assumed that antibiotic use promotes conjugative transfer, evidence suggests this effect may be overestimated (44). Conjugation efficiencies between *E. coli* and the genetically tractable yeast species *Saccharomyces cerevisiae* are typically below 1 in 1,000 yeast cells (45) (vs. ~1 in 100 efficiency per cell for 1 µg DNA in a LiAc transformation

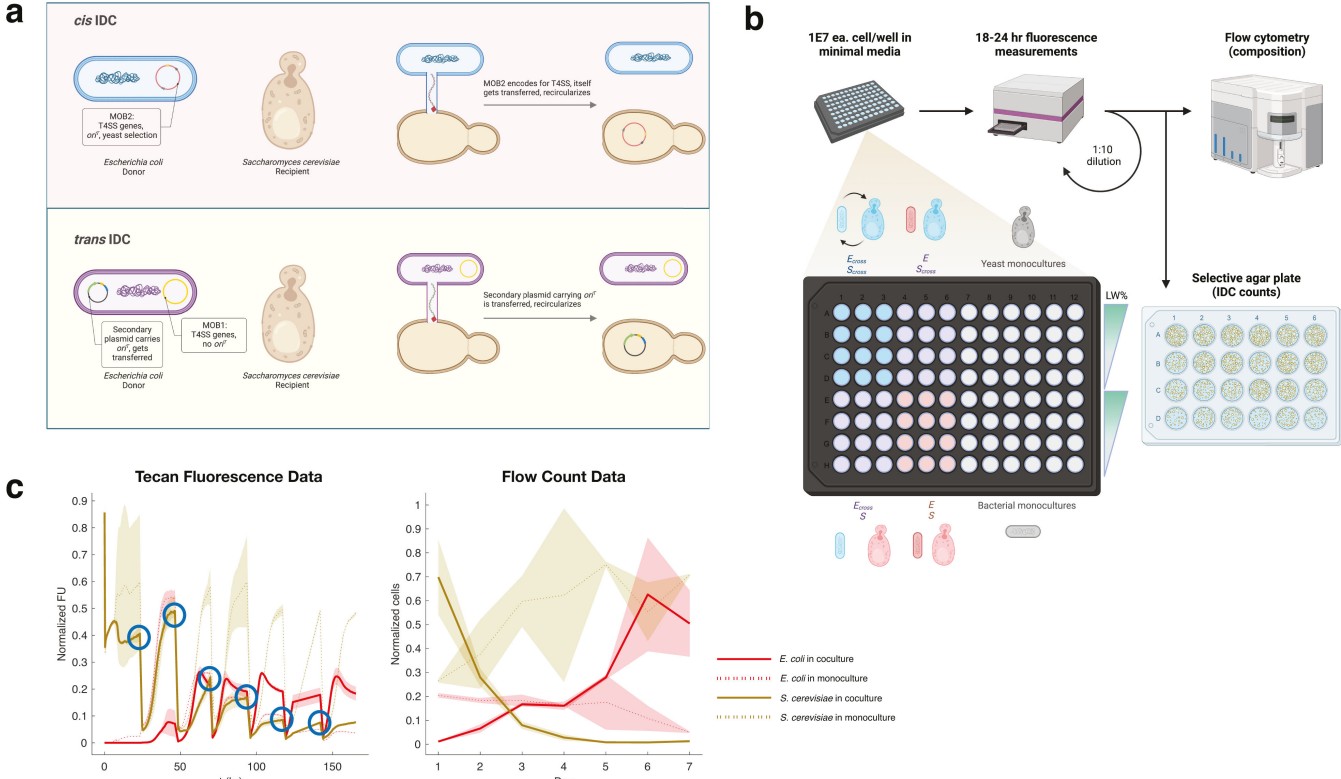

**FIG 1** Setup of IDC cultures with metabolic crossfeeding. (a) Types of IDC transfer. For this IncP T4SS system, DNA transfer to recipient *S. cerevisiae* can either occur in *cis* or in *trans*. For *cis*-IDC, the plasmid encoding the T4SS (pTA-Mob 2.0) also contains *S. cerevisiae* centromeric DNA maintenance machinery, selection genes HIS3 and URA3, and the transfer recognition sequence, oriT, at which the relaxosome nicks plasmid DNA and transfers it through the pilus to the recipient. In *trans*-IDC, the T4SS-encoding plasmid (pTA-Mob 1.0) lacks the sequences for *S. cerevisiae* maintenance, *S. cerevisiae* selection, and the oriT sequence. Thus, a second plasmid is required for *trans*-IDC, which carries these elements and is transferred to recipients. (b) Experimental setup of batch cultures. Cells are combined in a 96-well microplate with varying levels of leucine (L) and tryptophan (W) in a minimal media cocktail. Cocultures and monocultures are incubated at 30°C with continuous shaking and measured for fluorescence of each species every 15 minutes. After 18–24 hours of growth, cells are diluted 1:10 into new media to continue growing. Simultaneously, a 1:10 dilution of cells is prepared for flow cytometry, and an undiluted 100 µL is plated onto IDC-selective plates. (c) Example of culture composition measurements. Traces of cells in culture, measured by either plate-reader fluorescence (left) or flow cytometry cell count (right), normalized by dividing each reading by the maximum for that measurement type after day 1. For example, mCherry (*E. coli*) measurements are all divided by the maximum mCherry reading for all samples during experiments after day 1. Cell counts are normalized per species, fluorescence per fluorophore. Red traces denote bacterial growth, and brown traces denote *S. cerevisiae* growth. Solid lines represent cell growth in coculture (mean of four replicates), dotted lines represent growth in monoculture (mean of two replicates). Shaded regions are standard deviation from mean. Blue circles in the fluorescence plot denote times at which samples are diluted for batch culturing and additional measurements. Example shown is crossfeeding pair ($E_{cross}$ $S_{cross}$) at 15% Leu and Trp (15% LW).

(46)), though recent work has succeeded in generating 10-fold higher efficiencies by selectively mutating the T4SS machinery (40). Synthetic approaches also exist for increasing efficiency, such as colocalizing donor and recipient cells on beads, but have limited utility outside of laboratory settings (47). Bacterial recipients of horizontal gene transfer can subsequently serve as conjugative donors, leading to an exponential increase in conjugated cells (47). In contrast, IDC recipients are unable to subsequently serve as donors. This is ideal for biocontainment but further limits efficiency. In the laboratory, it is easy to overcome low conjugation efficiencies by selecting specifically for transconjugants and subculturing them. However, whether IDC can be optimized to modify enough individuals to affect population-level outcomes, as would be required for *in situ* modifications or control of engineered consortia, has not been tested.

In this work, we explore strategies for controlling IDC to affect population-level outcomes. One such strategy is to tune donor and recipient population ratios, since IDC rates are considered to be dependent on the frequency of donor-recipient interactions, and some work has demonstrated an increase in IDC for higher donor-to-recipient

ratios, albeit on shorter time scales (45). We thus use strains of *E. coli* and *S. cerevisiae* mutated to allow tunable population control via engineered crossfeeding between *E. coli* and *S. cerevisiae*, in which each species is auxotrophic for an essential amino acid that the other species overproduces. This approach also has implications in colony settings, which more closely match the dense biofilm environments in which most microbes naturally exist (48). Here, conjugation events between two populations occur along population boundaries (49), and mutualism between cells can greatly increase intermixing of populations in both bacteria (50) and yeast (51), hypothetically creating more population boundaries along which IDC can occur. We also probe the spatial dynamics of supposedly well-mixed cultures, since *E. coli* and *S. cerevisiae* are known to form mixed cellular aggregates via mannoprotein binding of *E. coli* (52), and determine how these short-range interactions affect both population dynamics and IDC. We model IDC in culture via a series of ordinary differential equations (ODEs), to predict both IDC transfer terms and conditions for which IDC is optimized. Finally, to verify whether these tunable population knobs can cause population-scale recipient changes via IDC, we apply them to control IDC rescuing and IDC killing of recipient yeast populations.

## RESULTS

### Crossfeeding mixed cultures and automation enable control and measurement of population ratios

To enable tunable control of population ratios, we designed strains of *E. coli* and *S. cerevisiae* to be obligate mutualists when deprived of specific nutrients. We utilized a yeast strain (51) that is auxotrophic for tryptophan (Trp⁻, Δ*trp2*, encoding anthranilate synthase, precursor to tryptophan synthesis [53]) and overproduces leucine (Leu⁺⁺, *LEU4^FBR* via feedback resistant mutation [54]), carrying a genomically integrated constitutive ymCitrine fluorescent reporter (*his3Δ::prACT1-ymCitrine-tADH::HIS3MX6*). We also developed a corresponding leucine-auxotrophic, tryptophan-overproducing crossfeeder *E. coli* (Leu⁻, Trp⁺⁺, via Δ*leuA*—responsible for leucine intermediate 2-isopropylmalate synthesis [55]—and Δ*trpR*, the canonical Trp repressor) that constitutively expresses mCherry episomally. Along with the corresponding strains of *S. cerevisiae* and *E. coli* unmodified for *LEU* and *TRP* (hereafter referred to as wild type[WT] strains "S" and "E," respectively), these "crossfeeder" strains ("S_cross" and "E_cross") allowed us to tune the concentrations of leucine and tryptophan in mixed culture to alter steady-state cell ratios.

To track the dynamics of each strain in mixed culture in addition to IDC counts over several days, we employed a multifaceted measurement scheme (Fig. 1b). Each cell pairing was batch cultured in a 96-well plate with various concentrations of leucine and tryptophan ("% LW") and measured for fluorescence of each strain (ymCitrine for yeast, mCherry for bacteria) in 15-minute intervals via an automated plate handler and a fluorimeter, yielding strain-specific dynamic information. After ~18–24 hours, batch cultures were diluted 10-fold into fresh media but, in most cases, were also measured via flow cytometry for verification of cell counts, allowing us to utilize the difference in cell sizes between bacteria and yeasts to optically measure individual cells of each type in a high-throughput way. This dual-measurement scheme allowed dynamic growth information while also controlling for variation in fluorescence expression due to, e.g., growth phase (Fig. 1c). Furthermore, we plated mixed cultures onto IDC-selective agar media after each day of batch culturing to get raw IDC counts. We measured population effects on IDC both in *cis*—with the self-transferring plasmid pTA-Mob 2.0—and in *trans*, via a two-plasmid system including the *ori^T*-lacking pTA-Mob 1.0 and a separate, yeast-selectable transfer plasmid (Fig. 1a). After screening for conditions that maximize crossfeeder growth while allowing comparable growth of each species in fully supplemented media (Fig. S1), we were able to batch culture cells in a range of strain-dependent leucine and tryptophan concentrations for at least 6 days. This integrated approach allows us to dynamically track batch cultures of bacteria and yeast, while also measuring cell counts for bacteria, yeast, and transconjugants each day.

## Tuning population ratios in batch culture affects the number of IDC events

To determine if we can control IDC frequencies by tuning steady-state population growth, we tested all combinations of crossfeeder *E. coli* ($E_{cross}$) and *S. cerevisiae* ($S_{cross}$) and their WT counterparts (E and S), over a range of leucine and tryptophan concentrations (% LW). In most cases, bacterial and yeast populations failed to establish stable crossfeeding with leucine and tryptophan fully removed from media—i.e., they were unable to support each others' growth without external sources of leucine and tryptophan—and interactions between species seemed primarily antagonistic, with one strain's growth corresponding to the loss of the other. Likely this is, at least in part, due to direct competition, since both species relied on the same sugar source (glucose), but other mechanisms are possible and we didn't experimentally validate the antagonistic mechanism. Crossfeeding bacteria temporarily survived via metabolites secreted by crossfeeding yeast (54) before the latter is outcompeted, driving down both populations; in contrast, auxotrophic yeasts did not benefit similarly from Trp-overproducing bacteria (56) (Fig. S2). Moreover, auxotrophic $E_{cross}$ bacteria survived from WT yeasts at 0% leucine and tryptophan (0% LW), while having no obvious effect on WT yeast growth, in an apparent commensal relationship, suggesting either a low but significant level of basal leucine (or synthesis intermediate) secretion from S via an unknown mechanism (57) or sufficient yeast lysate for $E_{cross}$ survival. WT bacteria ("E") did not provide a similar benefit for crossfeeding yeasts (Fig. 2a; Fig. S2).

As in a previous work, we found markedly lower *trans* IDC rates relative to *cis* IDC (Fig. S3) (58). Contrary to previous work demonstrating higher IDC rates with more donor bacteria (45), however, we found an inverse correlation between donor-to-recipient ratios and IDC counts over time (Fig. 2b), especially for *cis* IDC (Fig. S3). That is, more donor bacteria resulted in less conjugated yeast. This trend manifested as a linear fit on a log-log plot, with an increasingly negative slope over time (Fig. 2c and d). Some features of this relationship were seemingly due to changes in recipient populations—e.g., $S_{cross}$ eventually died off when paired with E, likely raising the ratio of donor bacteria to recipient yeast (labeled "D:R") and lowering the IDC (blue dots, Fig. 2c). However, if yeast density changes were the only cause of the inverse relationship in Fig. 2c, we'd have expected the IDC-per-recipient frequency to be constant for all pairings, but we found variations between strains (Fig. S4), suggesting that other factors could also be at work. These findings suggest that, despite the lack of stable crossfeeding at 0% LW, we can still control IDC by tuning populations, since steady-state ratios of donors-to-recipients are inversely correlated to IDC counts.

## Mannoprotein-based cell adhesion mediates IDC and affects bacterial commensalism

Since IDC depends on cell-cell collisions in culture, we explored how known adherence mechanisms between *E. coli* and *S. cerevisiae* affect IDC. Mannoproteins are ubiquitous in fungal cell walls (59), and type I fimbriae in *E. coli* bind to these (60, 61), forming bacteria-yeast "clumps" that can affect crossfeeding dynamics (52). We thus repeated our batch culture experiments for population dynamics and IDC with- and without mannose added to the media, which saturates bacterial mannose receptors and reduces clumping. These cultures were measured dynamically for fluorescence as per previous experiments (Fig. 1b) but here were also imaged via fluorescence microscopy to optically assess the extent of clumping.

Fluorescence microscopy analysis replicated previous findings (52) showing that adding mannose to growth media prevented most bacteria-yeast clumping (Fig. 3a). Image analysis demonstrated that the size of yeast clumps—a proxy for number of yeast cells per clump—increased concurrent with the number of bacteria in a clump ("coincident bacteria"), implying that bacteria mediate cell clump formation (Fig. S5 and S6). Interestingly, we found that mannose-infused media prevented nearly all IDC, with only a few samples yielding single-digit IDC counts by the end of a 6-day time course, roughly 10-fold fewer than corresponding samples without mannose (Fig. 3b). Moreover,

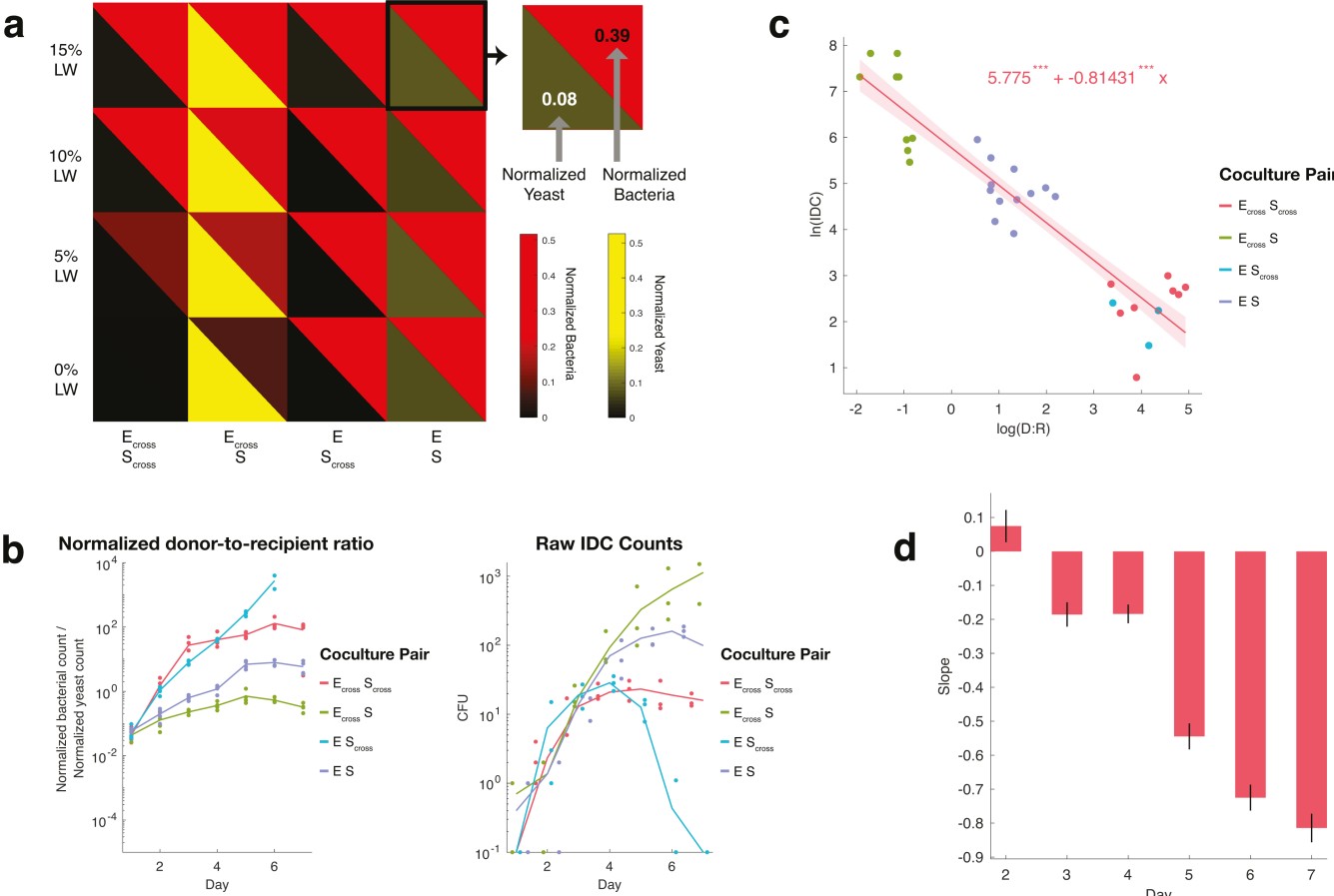

**FIG 2** Batch culturing crossfeeding populations reveals relationships between population ratios and IDC. (a) Compositional outcomes of cocultures. Split heatmap of *E. coli* cell counts (red) and *S. cerevisiae* cell counts (yellow) for each coculture pairing (columns) over a range of LW concentrations (rows), from flow cytometry of batch culture day 6 (mean of four replicates). Counts are normalized to max cell counts per species and then multiplied uniformly to enhance color brightness, to better visualize low-growing populations. At 0% LW, crossfeeding pairs' ($E_{cross} S_{cross}$) growth is imperceptibly small, while the crossfeeder *E. coli* paired with WT *S. cerevisiae* ($E_{cross}$ S) shows bacterial commensalism, and the WT pair (E S) shows a basal interaction with relatively higher bacteria and lower yeast counts. Experimental results are for *cis*-donors. Brightness is mean of four replicates. (b) Normalized donor-to-recipient ratios correspond inversely to IDC counts. Ratios of normalized cell counts (cell count divided by maximum for that species across experiment) of *E. coli* donors and *S. cerevisiae* recipients, calculated from flow cytometry data, plotted over time for each cell pairing, at 10% LW (left). Raw IDC counts from colony forming units on selectable media for the same conditions and cell pairings (right). Counts ≥ 200 should be assumed estimates. Lines represent means of four replicates (dots), colors, and coculture pairings. Inset: determination of donor-to-recipient ratio ("D:R"). (c) Correlation between donor-to-recipient ratios and IDC is linearly inverse on log-log scale. Log-log distribution of donor-to-recipient ratio and IDC counts for each cell pairing at day 7, for all % LW. IDC counts at detection limits (500 or 0.1 per 100 μL sampling, see Materials and Methods) omitted; counts ≥ 200 are estimates. Generalized linear model fit, with normal distribution, shown in solid red line, with 95% CI in shaded region. Fit equation displayed with stars denoting *P* value significance for each term. (d) Inverse correlation between D:R and IDC increases over time. Slopes of log-log plot linear fits for all days of batch culture, showing decreasing slope over time. Black bars = standard error of mean.

mannose-supplemented samples showed fundamentally altered dynamics for $E_{cross}$-S pairings, with auxotrophic $E_{cross}$ cells unable to survive at 0% leucine and with much lower growth at higher percentages of leucine relative to mannose-free samples (Fig. 3c; Fig. S7). Thus, mannose interrupted the commensal dynamics previously seen without mannose.

## Deterministic models reveal boundaries of population control of IDC

To explore how the "knobs" of our system could be tuned to best affect population ratios and IDC and to better understand the differences between clumping and non-clumping populations, we used a set of ODEs to deterministically model our experimental conditions (see supplemental discussion for more details), based on previous work

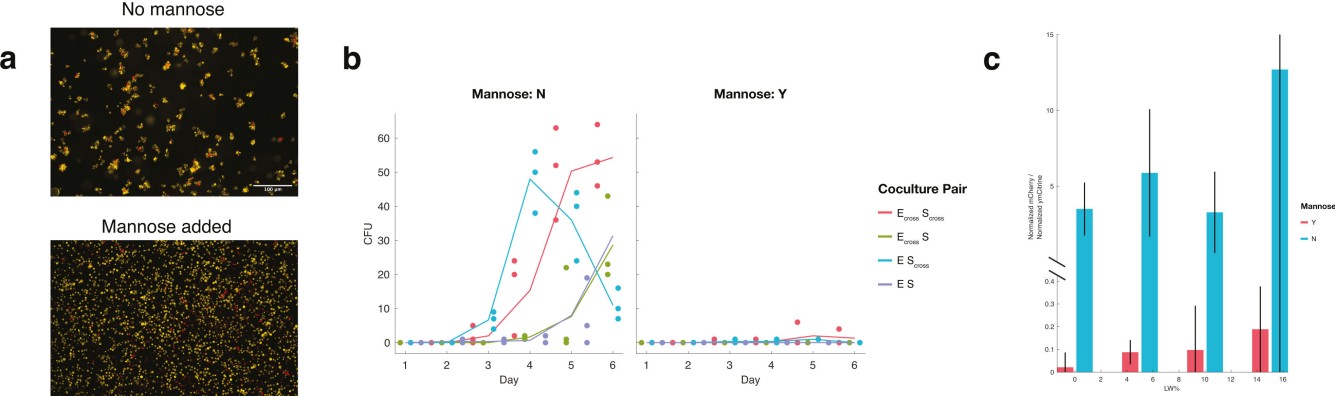

**FIG 3** Mannose disruption of cell aggregates lowers IDC and interrupts bacterial commensalism. (a) Mannose interrupts mixed aggregates in culture. Microscopy images of batch coculture after 6 days, either without (left) or with (right) mannose supplemented in media. Cells shown are *trans*-WT *E. coli* (E) and crossfeeding *S. cerevisiae* (S$_{cross}$) at 15% LW, chosen to exemplify differences in clumping. Samples were diluted 1:10, imaged with a 10× objective. Scale bar = 100 μm. (b) Interrupting clumps with mannose depresses IDC. Raw IDC counts (CFU) for samples at 15% LW without mannose supplementation (left column) are ≥ 10× those with mannose supplementation (right column). Lines represent means of three replicates, colors, and coculture pairings. (c) Interrupting clumps with mannose prevents commensalism for crossfeeding *E. coli*. Normalized donor-to-recipient ratios for commensal E$_{cross}$-S pairing across four different LW concentrations, after 6 days of batch culturing, calculated from fluorescence data. Mannose-minus (blue) and mannose-plus (red) samples show that clumping is required to sustain crossfeeding *E. coli* with WT *S. cerevisiae*, especially at lower % LW. Bars represent mean of three replicates, error bars, 95% CI, significance via two-sample *t*-test. 0% LW: $P = 0.0010$, $t = 8.543$, df = 4. 5% LW: $P = 0.0041$, $t = 5.9229$. 10% LW: $P = 0.0071$, $t = 5.0709$. 15% LW: $P = 0.018$, $t = 3.8773$. Degrees of freedom = 4 for all % LW.

modeling crossfeeding cocultures (62, 63). Moreover, to the best of our knowledge, no IDC transfer terms have been reported in terms of the ODEs for conjugative transfer first developed by Levin et al. (64) nor have any such models accounted for spatial heterogeneity in culture. To reveal the boundaries of IDC control, we first fit the results from mannose-supplemented experiments to a system of two ODEs representing total bacteria and total yeast (including transconjugants). We ran Latin Hypercube Sampling iteratively to randomly sample all parameters within a predicted range and calculated the total error between model outcomes and fluorescence data for bacteria and yeast. We then used this error to rank model parameters, which we adjusted and reran until key experimental results were demonstrated for each cell pairing, namely, susceptibility to amino acid supplementation, steady-state survival, and approximate donor-to-recipient ratio (see supplemental discussion, Fig. S8 and S9).

Once we found best-fit approximations of parameters in bacterial and yeast growth equations, we tested a wide range of IDC transfer terms $\gamma$ against data from mannose-supplemented experiments. This transfer term represents the fractional occurrence of conjugative transfer per, in this case, bacteria-yeast collision and has been previously approximated at $4*10^{-3}$ using a similar model for intraspecies transfer among enteric bovine *E. coli* (65) (Fig. 4a). We found, however, that $\gamma$ would have to be significantly lower, roughly between $7*10^{-6}$ and $4*10^{-5}$, to recapitulate our results in media containing mannose (Fig. 4b; Fig. S9).

We performed another round of parameterization against measurements of clumped cells growing in mannose-free media, using ODEs modified to include clumping. In this model, IDC was represented by two different transfer terms: $\gamma$ for free-cell collisions, as per previous fits, and $\gamma_c$ for clumped cells. The model fits (Fig. S10) yielded two possibilities that recapitulated the data: low $\gamma_c$ with $\gamma$ in the range of $5*10^{-4} - 1*10^{-3}$—higher than $\gamma$ values found in the free-cell model, thus probably not representative—or low $\gamma$ with $\gamma_c$ in the range of $2*10^{-4} - 4*10^{-4}$ (Fig. 4c; Fig. S11), only ~10-fold lower than the conjugation transfer term predicted for intraspecies transfer. Together, these results predict that conjugation transfer terms for IDC in ODE models are near those found for intraspecies transfer, but only for cells that are clumped. Importantly, we've strayed from much of the literature regarding IDC, which considers the transfer rate as a percentage of recipient

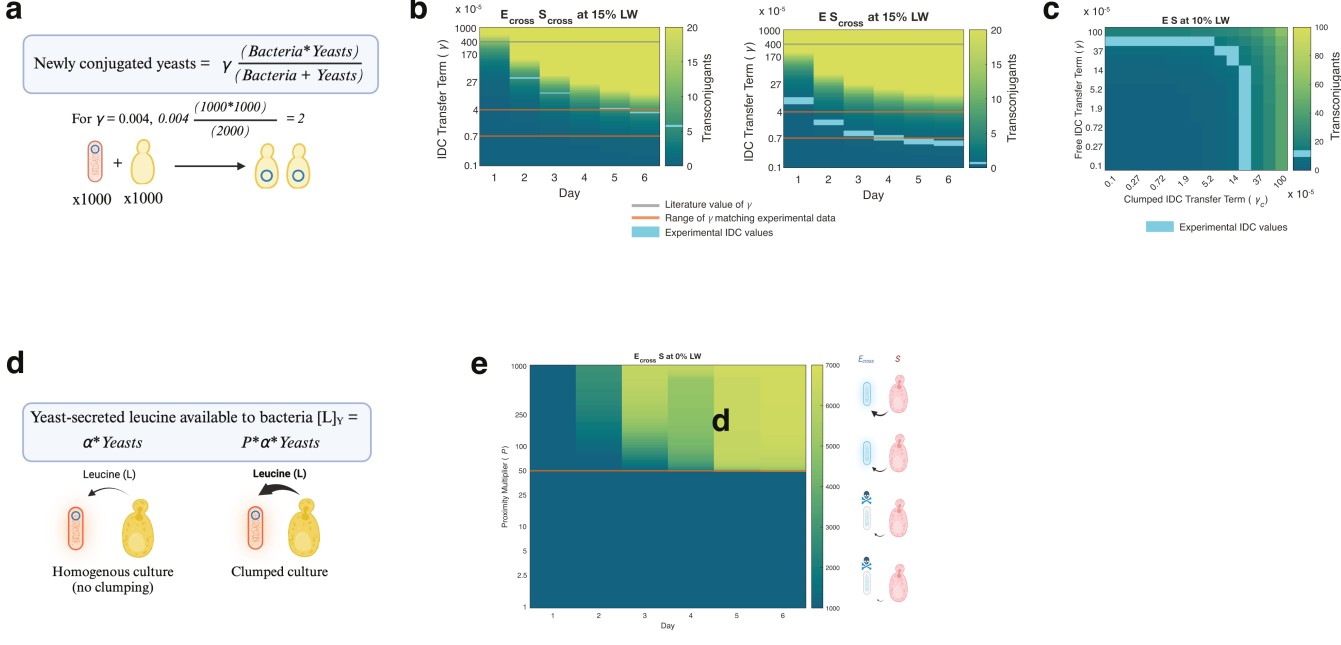

**FIG 4** Deterministic modeling shows limits of IDC transfer terms, proximity benefits, and in and out of aggregates. (a) Depiction of IDC transfer term $\gamma$. Schematic of how the IDC transfer term $\gamma$ relates to the accumulation of transconjugants. Idealized scenario of 1,000 bacterial (*E. coli*) and 1,000 yeast (*S. cerevisiae*) cells, using literature prediction of $\gamma = 0.004$ results in two new transconjugants due to conjugative transfer. (b) IDC transfer term sweep for "free" cell model. Heatmaps showing predicted number of transconjugants (heatmap) for a range of IDC transfer terms $\gamma$ (*y*-axis) over 6 days of batch culturing, assuming cells are unable to clump and thus conjugate via random collisions. Two of the four experimental conditions that yielded IDC counts > 0 with mannose are shown (mean experimental CFU of three replicates = cyan heat markers). Gray line at $\gamma = 0.004$ represents literature prediction for enteric *E. coli* conjugation transfer term[65], and orange lines represent range of IDC-transfer term values matching experimental data, roughly between $7*10^{-6}$ and $4*10^{-5}$. (c) IDC transfer term sweep for "clumped" cell model. Heatmap showing predicted number of transconjugants (color) for a range of "free" IDC transfer terms $\gamma$ (*y*-axis) and "clumped" IDC transfer terms $\gamma_c$ (*x*-axis), for one representative experimental condition at day 6 of batch culturing. Cyan boxes represent mean experimental IDC count of 18.7 per 100 µL culture (three replicates), with several values of $\gamma$ and $\gamma_c$ resulting in this number of transconjugants. Two main conditions yield the experimental IDC results: low $\gamma_c$ with $\gamma$ above $5*10^{-4}$ or low $\gamma$ with $\gamma_c$ near $3*10^{-4}$. Because free-model results showed $\gamma$ below $5*10^{-5}$, it's likely that the latter case is true, with most conjugation resulting from clumped interactions. (d) Depiction of proximity term $P$. In homogeneous (non-clumping) cultures, the concentration of leucine available to leucine-auxotrophic bacteria (*E. coli*), due to yeast (*S. cerevisiae*) secretion, is determined by a constant secreted molarity of leucine ($\alpha$) per yeast cell. In clumped cultures, we include a proximity multiplier $P$ that increases leucine available to bacteria, irrespective of the basal-secreted leucine per yeast. (e) Proximity term sweep for "clumped" model. Heatmap showing predicted *E. coli* fluorescence signal (color) for range of proximity-benefit multiplier $P$ (*y*-axis) over 6 days, for $E_{cross}$-S pairing at 0% LW. Orange bar shows approximate $P$ value matching experimental coculture data, i.e., a $P$ value high enough (~50) to allow $E_{cross}$ growth solely from clumping to WT *S. cerevisiae*. Based on the model structure, this implies that nutrient-dependent *E. coli* see ~50× benefit from proximity to WT *S. cerevisiae* in terms of nutrient access from yeast cells, though other mechanisms for this commensalism are possible.

cells (40, 45, 58, 66), though some examples of quantifying conjugation as a rate of cell coincidence exist, similar to the modeling we've done here (64, 67, 68). This is likely due to IDC's use as a substitute for DNA transformation, but for many potential applications, the per-donor IDC rate may be equally relevant, especially if donors are utilized as a temporary probiotic.

Additionally, a proximity term $P$ was used in this model to account for changes in benefit arising from the proximity of clumped cells, which allows for $E_{cross}$ survival with WT yeasts (S) (Fig. 4d). $P$ value sweeps showed an apparent amino-acid secretion increase on the order of 50× from WT yeasts (S), to allow $E_{cross}$ cells to grow in 0% leucine (Fig. 4e). While $P$ mathematically served to multiply the amino acid secretion term in the model, it could just as plausibly have resulted from leucine (intermediate) in yeast cell lysate or some other mechanism of bacterial benefit. Both explanations had caveats, however. While it might make sense to assume that the increase in $P$ is due to greater proximity to yeast with low nutrient secretion, there's no evidence that WT yeasts secrete leucine (intermediates) basally (57). On the other hand, a yeast-lysate

explanation disagrees with our observed higher IDC counts for these samples, corresponding to yeast that continued to grow after existing in such a clumped state, wherein they acquired the conjugated DNA. Moreover, this model didn't deconvolute whether mannoprotein binding led to both higher IDC and commensalism independently or only higher commensalism which in turn increased IDC. Still, the model served both to add to our knowledge of how key features of this system function and allowed us to predict IDC outcomes for various experimental parameters, as we explore later.

## Mixed colonies exhibit inverse donor-to-recipient to IDC relationship and low spatial intermixing

Having thus far characterized IDC in well-mixed liquid cocultures, we next sought to understand how population dynamics affect IDC in spatially constrained settings, to better predict IDC functionality in natural settings such as biofilms (48). We emulated "expansion" assays (69), which have previously demonstrated greater intermixing of mutualistic populations (50, 51) by repeating batch culture initial conditions on 2% agar minimal media plates, except with 10-fold fewer initial cells. We pipetted ≥18 2 µL mixed-cell droplets onto plates and allowed them to grow continuously for 6 days. We imaged three colonies for 2D spatial distribution each day via wide-field fluorescence microscopy and another three that were then scraped, washed, and diluted for composition and IDC measurements (Fig. 5a and b).

As with batch cultures, there was an inverse correlation between donor-recipient ratios and IDC in most cases, though with greater noise (Fig. S3). However, IDC-per-recipient rates remained relatively constant, unlike cultures (Fig. S4). These differences from culture conditions might be due to "jackpot" populations, in which a genetic island of transconjugants finds a spatial niche among the stochastic colony front (70), resulting in a wider range of IDC counts for each condition (see supplemental discussion, Fig. S13). Because conjugation has been shown to occur along population boundaries (49), we determined relative population mixing by calculating colocalization (71) of bacterial and yeast fluorescence signals (see supplemental discussion). While colocalization did positively correlate with overall IDC values, most mixed colonies had very low colocalization, suggesting once more that antagonism dominates population dynamics between these species (Fig. 5c and d).

## Population dynamics can be tuned to rescue a recipient population through IDC

To test whether population control of IDC can be used to alter recipients at a population scale (we know it can alter individual recipients), we next sought to "rescue" starved yeast cells with poor or non-existent growth, via genes carried on the transferred DNA (Fig. 6a). We first tested this with the *cis*-IDC plasmid pTA-Mob 2.0, which carries *HIS3* and *URA3* and allows transconjugants to grow in media deficient for uracil and histidine. IDC from WT donors mostly failed to rescue U or H-auxotrophic yeast recipients growing in low concentrations of uracil and histidine (% UH), as the bacteria competed the yeasts to collapse before sufficient transconjugant growth could establish (Fig. S14).

Our previous results showed higher IDC for lower donor-to-recipient ratios, so to increase the likelihood of rescue, we used auxotrophic bacterial donors at 0% leucine. These donors can thus only survive if the paired yeasts metabolically support them. Remarkably, we found a drastic increase in IDC rescue from $E_{cross}$ donors, for both $S_{cross}$ and S recipients, an effect that varied by uracil and histidine amounts (Fig. 6b and c; Fig. S14). $E_{cross}$ rescued both recipient strains with greater speed and efficiency than E did in all cases, though at 0% UH, paired crossfeeder populations collapsed (Fig. 6c). At intermediate concentrations of uracil and histidine—especially 5% UH—rescue showed high stochasticity, as some biological replicates were fully rescued while others collapsed (Fig. S15). We also used our clumping model to predict the range of possible rescue outcomes for each cell pairing over a range of amino acid concentrations (Fig. 6d; Fig. S16). With minimal alterations to account for experimental differences, the model

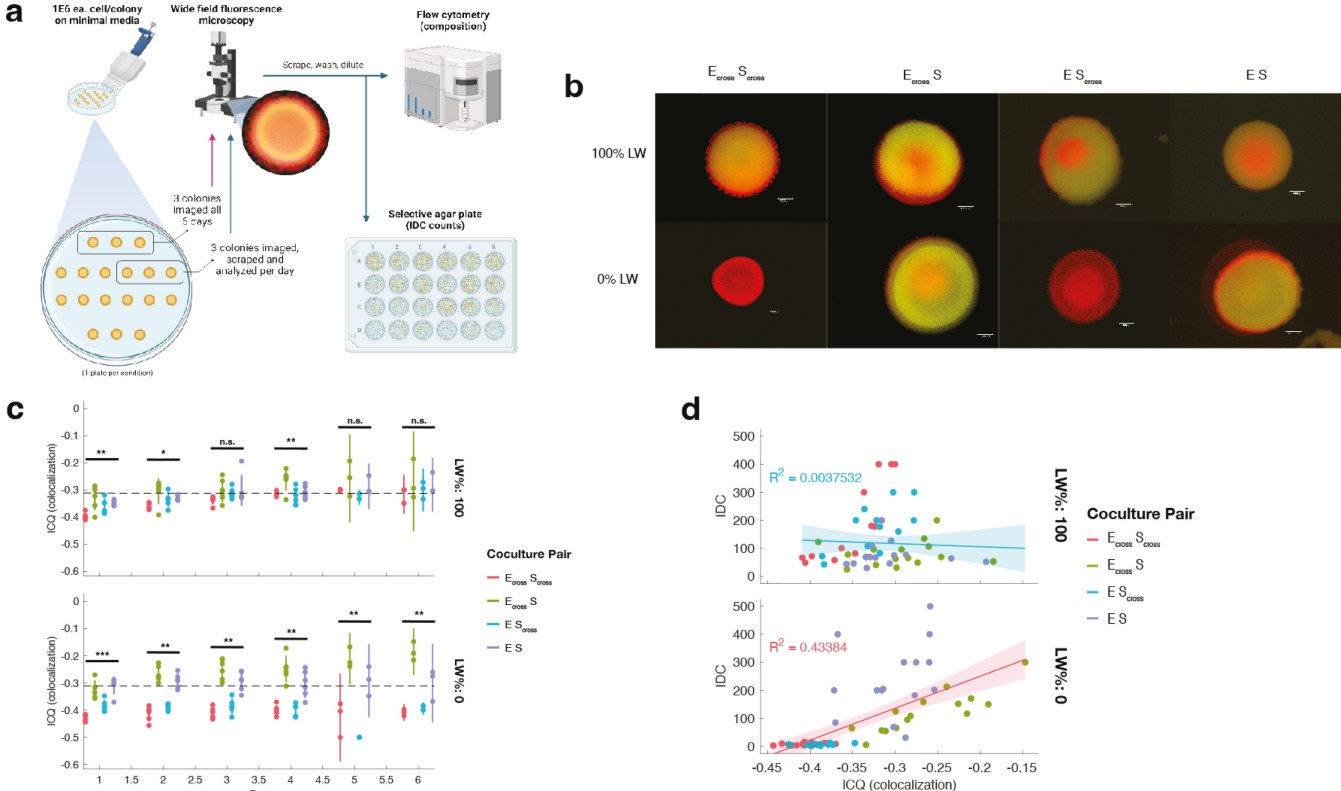

**FIG 5** Mixed colonies follow similar dynamics to culture conditions and show increased IDC for more spatially mixed populations. (a) Experimental setup of colony assay. Cells are combined and pipetted onto minimal media with 2% agar. Each plate contains ≥18 colony replicates of one cell pairing, and one amino acid concentration. After each day's growth, six colonies are imaged with a wide-field fluorescence microscope. Three of these continue to be imaged daily, while the other three are scraped, washed, and diluted for flow cytometry and IDC plating. (b) Example of mixed colony cell distribution. Fluorescence microscopy images for each cell pairing at 100% LW (top row) and 0% LW (bottom row). All pairings here include *cis*-donors. *S. cerevisiae* are displayed in yellow channel, *E. coli* in red. Channels are scaled for brightness to emphasize distribution, scale bar = 1,000 μm. (c) Colocalization shows divergent intermixing at 0% amino acids, though antagonism drives spatial distribution overall. Li colocalization analyses of colonies (intensity correlation quotient [ICQ] = 0.5 is complete colocalization between channels, ICQ = −0.5 is complete spatial segregation) show range of outcomes for 0% LW colonies, less so for 100% LW colonies. All experimental ICQ values range from −0.1 to −0.5, implying antagonistic spatial segregation. Distributions shown are of *cis*-donor pairings. Calculated ICQ values for each replicate and condition represented by dots, 95% CI of the mean by vertical bars. Stars denote *P* values from ANOVA one-way test of 95% confidence interval between all four pairings at each day and % LW using sum of squares test (see Table S6 for *F*-values and degrees of freedom). Two-sample *t*-tests of day 6 ICQs (not displayed) show significant differences between 0% LW and 100% LW for $E_{cross}$-$S_{cross}$ pairing ($P = 0.0064$, $t = −5.2247$, df = 4) and E-$S_{cross}$ pairing ($P = 0.0073$, $t = −5.0285$, df = 4), while $E_{cross}$-S and E-S pairings do not show differences between LW% ($P = 0.15$, $t = 1.770$, df = 4 and $P = 0.64$, $t = −0.5091$, df = 4, respectively). (d) Colocalization correlates positively with IDC values. ICQ values plotted against raw IDC (CFU) counts for *cis*-donor pairings, at 0% and 100% LW, for IDC ≥ 2. For the smaller range of ICQ values at 100% LW, IDC counts slow little divergence, whereas at 0% LW, IDC correlates positively with ICQ. IDC counts ≥ 200 are estimates.

recapitulated our experimental results: for most concentrations of U and H, and with L concentration kept low, bacterial antagonism is minimized, and greater IDC is possible, allowing for the increased rescue of yeast seen in these experiments (Fig. 6d and e; see supplemental discussion for model changes).

## IDC-mediated CRISPR killing can be interrupted by mannose addition

We next tested whether we could collapse or depress a recipient yeast population via IDC-mediated killing. We designed a conjugatable CRISPR/Cas9 system that can be transferred from bacteria to yeast, where it targets a BFP and *URA3*-carrying plasmid in recipients, such that destruction of this plasmid would render recipient cells unable to grow in uracil-deficient media. Unlike most Cas9 systems, which utilize a repair sequence to replace the cut DNA, we relied on repeated cutting of the target DNA with no repair,

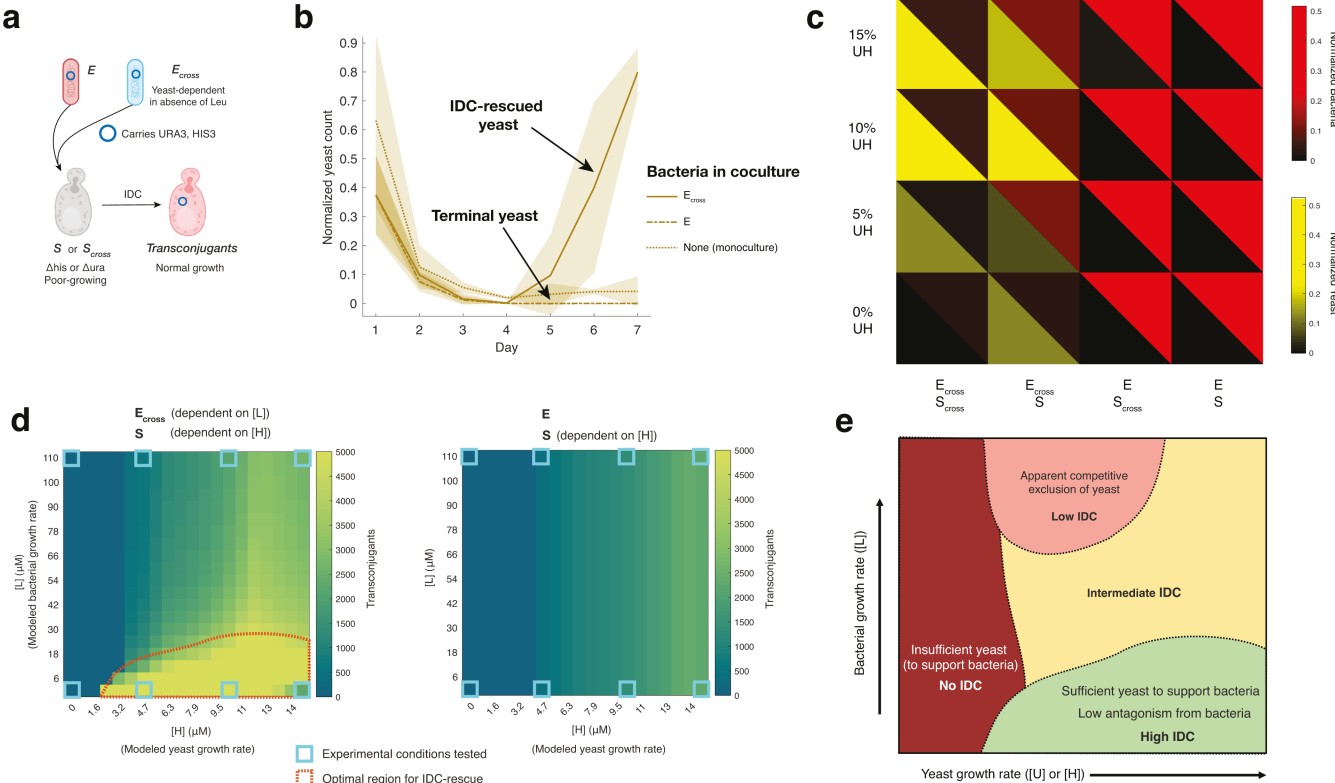

**FIG 6** Utilizing population dynamics allows IDC-mediated rescue of unhealthy recipient populations. (a) Rescue setup: *S. cerevisiae* strains auxotrophic for either uracil (U) or histidine (H) are grown in low [U] or [H], but IDC restores prototrophy in recipients, allowing them to grow normally. Rescue was tested with both WT donors (E) or leucine auxotrophs (E$_{cross}$) at 0% L, to test effect of population tuning. (b) *S. cerevisiae* growth is rescued by crossfeeding donor IDC. *S. cerevisiae* cell counts from flow cytometry, normalized to max count after day 1, for each cell pairing of S$_{cross}$ at 10% UH, 0% L. In monoculture (dotted line), S$_{cross}$ grows poorly at 10% UH. WT *E. coli* donors antagonistically depress S$_{cross}$ growth (dot dash line), despite their ability to transfer IDC plasmid that would rescue recipients. E$_{cross}$ donors, on the other hand, are able to transfer sufficient rescuing plasmid (solid line), allowing full S$_{cross}$ rescue. Means of six replicates over two experiments shown as traces, shading as standard deviation. (c) Batch culture growth in rescue assay shows greater success for starved donors across several conditions. Split heatmaps of normalized cell counts from flow cytometry for four cell pairings (columns) and four concentrations of uracil and histidine (rows). All samples grown with 0% leucine to starve E$_{cross}$ (*E. coli* in red). *S. cerevisiae* (yellow) auxotrophic for URA3 or HIS3 show greater growth upon receiving conjugated pTA-Mob 2.0 (*cis*), which is only significant when paired with E$_{cross}$. Brightness is mean of six replicates across two experiments, normalized to max cell count per species and experiment, multiplied uniformly to visualize low-growing strains. (d) Model prediction of rescue phase map. Select predictions for IDC counts based on clump model, adapted for rescue assay conditions. Concentrations of L (*y*-axis) and H (*x*-axis), for WT *S. cerevisiae* (S) cell pairings shown, with experimental amino acid values highlighted with cyan boxes. Note that while experimental [L] only includes 0% L and 100% L, the model predicts a range of [L] over which E$_{cross}$ could rescue *S. cerevisiae* more effectively than WT *E. coli* (E). (e) Conceptual phase map of IDC outcomes. Comparing growth of bacteria (*E. coli*, *y*-axis) and yeast (*S. cerevisiae*, *x*-axis), as controlled in rescue assay by amino acid levels. At low-enough growth for both species, populations collapse before sufficient IDC can occur. When bacteria are sufficiently supplied with nutrients (or aren't dependent on them), antagonism suppresses yeast growth, limiting rescue capacity. At low bacterial fitness, but moderately low yeast fitness, enough yeast cells are present to sustain growth of the starved bacteria for long enough to allow IDC and the lack of antagonism from E$_{cross}$ donors allows for optimal rescue of recipient population.

since our goal was simply to suppress the target cells' growth. Targeting an episomal sequence was also essential, to discern both IDC rates and cutting efficiency separately, without the lethality of cutting genomic DNA in recipient yeast (Fig. 7a). After verifying that the IDC-Cas9 plasmid is efficient for cutting its target via both direct yeast transformation and IDC (Fig. S17), we batch cultured crossfeeding yeast (W auxotrophs) containing the *BFP-URA3* plasmid at low levels of tryptophan and 0% uracil, along with donor cells that contained either a functional IDC-Cas9 system or one lacking an *ori^T* sequence and thus unable to transfer DNA. At 1% W, all yeast cultures died out, while at higher levels of W (5% and 10%), bacterial antagonism resulted in depressed yeast levels relative to monoculture yeast growth (Fig. 7b; Fig. S18). Donors carrying IDC-Cas9 ("cutters")

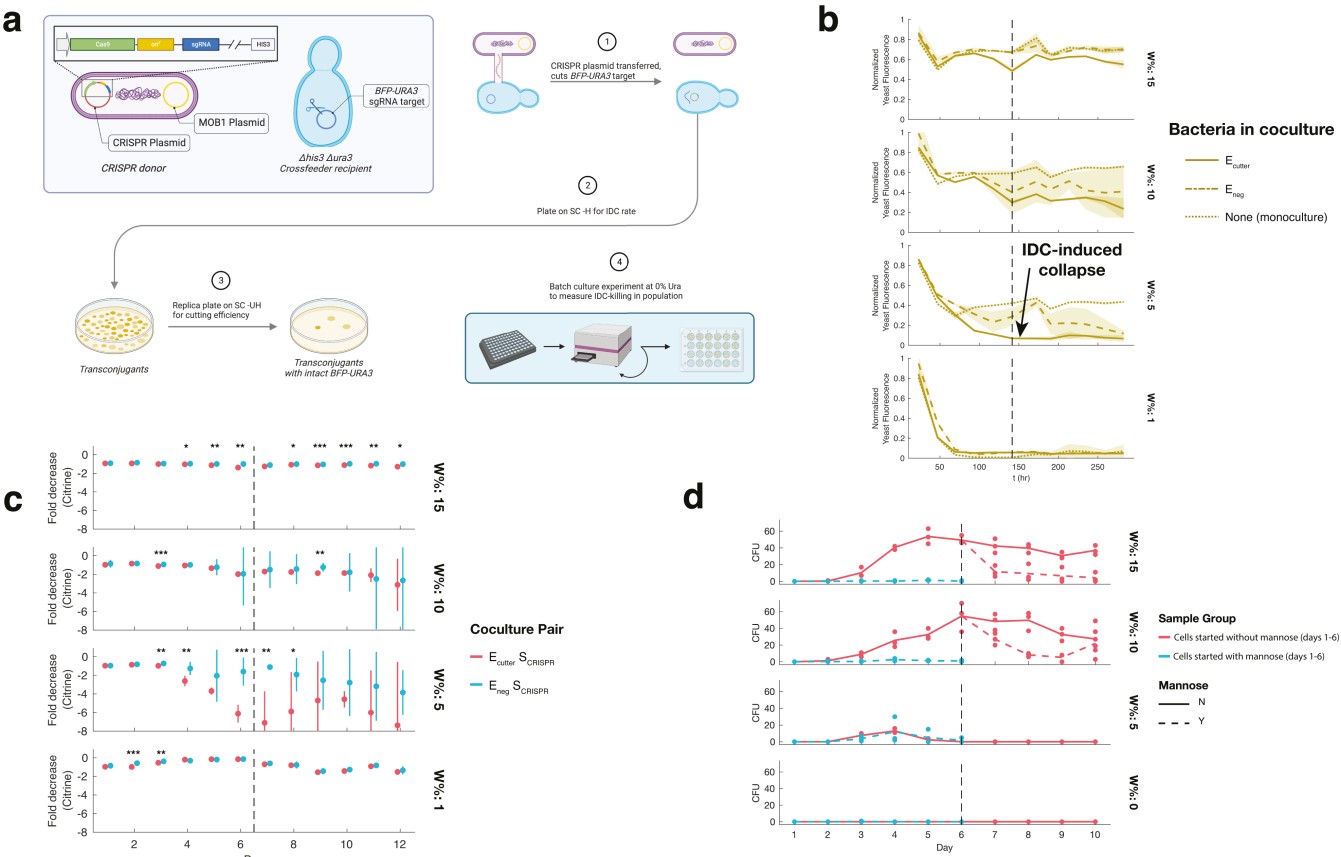

**FIG 7** IDC-mediated CRISPR killing is able to drive recipient population extinction and is mannose-interruptible. (a) Design of IDC-mediated CRISPR system. pTA-Mob 1.0 T4SS plasmid (*trans*) is paired with a Cas9 plasmid that contains the *ori$^T$* sequence (allowing for transfer), HIS3 yeast selection marker, and sgRNA coding for a connector region in BFP-URA plasmid. Recipient *S. cerevisiae* are *Δura30* and *Δhis3::HPHMX6* and carry BFP-URA plasmid. Upon IDC transfer, BFP-URA plasmid is cut via Cas9, with no repair template, but *S. cerevisiae* can continue to grow in media supplemented with uracil. In this way, we can measure IDC efficiency independently from CRISPR cutting efficiency, by plating for IDC (SC-H) and then replica plating for cut *S. cerevisiae* (SC-UH). Finally, cut-verified donors are grown in batch culture with CRISPR recipient yeast at 0% U to gauge ability to depress recipient population through IDC killing. (b) Growth plots of cocultures show IDC killing in some conditions. Fluorescence measurements shown here at the end of each day, from 12 days of batch culturing of cell pairs at four concentrations of tryptophan (rows). Trp-auxotrophic recipients ($S_{cross}$) collapse for both pairings (cutting donor—solid line—and no-*ori$^T$* negative control donor—dashed line) at 1% W, but only for the cutting donor at 5% W. Lines are means of three replicates, and shaded region represents standard deviation. Mannose was added to experiment after day 6, to break up cell clumps, shown as vertical dotted lines. (c) Comparing recipient population decline between cocultures and monoculture. Fold decreases in *S. cerevisiae* growth, based on normalized fluorescence, comparing *S. cerevisiae* in coculture to monoculture. Fold decrease = −(normalized monoculture citrine)/(normalized coculture citrine). Points are means of three replicates, bars 95% CI. Stars represent *P* value significance from two-sample *t*-test, with no significance for time points lacking stars (*P* > 0.05). Vertical dotted lines designate addition of mannose at day 6, to break up cell clumps. (d) Mannose prevents and reverses IDC. IDC counts (CFU) for *S. cerevisiae* cocultured with CRISPR-donating *E. coli* (negative control donor is unable to transfer DNA, all counts = 0, not shown), over four W% (rows). For days 1–6, half of cocultures were grown in mannose (blue dash, mean of three replicates), preventing most IDC relative to samples grown without mannose (red solid, mean of three replicates). After day 6 (vertical dotted line), non-mannose samples were split into media containing mannose (red dash, mean of three replicates) and media without it; surviving cocultures drop in IDC counts after mannose addition. Results are from repeat of experiment shown in parts b and c. IDC counts here are "transient" because transconjugants are terminal at 0% U, so transconjugants are unable to persist in coculture across days.

significantly depressed yeast growth beyond antagonism-based decreases, especially at 5% W, where yeast growth was decreased several fold beyond non-transferring control donors (Fig. 7c).

To gauge whether any effects of IDC could be reversed by interrupting cell clumps, we switched batch cultures to mannose-supplemented media after 6 days of growth and allowed them to grow for another 6 days. In both coculture pairings, subsequent yeast growth stopped declining after the media switch and persisted at steady-state levels

from day 6, ending trends of decline in both coculture pairings, though never recovering recipients completely to previous (higher) levels (Fig. 7b; Fig. S18). From this data, we were thus able to discern the extent to which recipient populations are depressed by antagonism from coculture with bacteria, versus IDC-mediated cutting, since the positive and negative donors are equivalent for fitness and ability to adhere and form pili to recipients (Fig. 7c; Fig. S19). The use of mannose allowed both prevention and reversal of IDC, with IDC dropping after day 6 for samples switched into mannose media (Fig. 7d). Since transconjugants cannot survive in 0% U, it should be noted that IDC counts are effectively transient "snapshots" of yeasts carrying IDC-Cas9 that have not yet been diluted out of batch culture or died from starvation.

## DISCUSSION

In this work, we developed tools to tune IDC and demonstrated their potential for perturbing yeast populations. We found the amount of IDC to be inversely correlated to the donor-to-recipient ratio in mixed cultures and colonies. This insight allowed us to demonstrate IDC rescue of a collapsing recipient population by tuning cocultures to minimize this donor-to-recipient ratio, driving up the transfer of an essential gene to recipient yeast. We also found that mannoprotein binding of bacterial donors to recipient yeast cell walls yielded approximately 10-fold higher IDC and that this binding could be precluded by adding mannose to growth media. We developed an IDC-mediated CRISPR/Cas system to kill recipient cells—achieving recipient population depression for some conditions and population collapse in one instance—and demonstrated the capacity of mannose to reverse this IDC-mediated perturbation. Finally, we determined that IDC correlates positively with spatial colocalization of cells in mixed colonies and that for this system, spatial ecology is primarily antagonistic, with each species preferentially growing away from the other species.

While our crossfeeding mutations didn't support steady-state obligate mutualism, auxotrophic *E. coli* were able to form commensal relationships with even WT *S. cerevisiae*, yielding some of the highest IDC values we observed. This presents an opportunity to implement conjugation systems in which only donors are engineered. This is true, too, for mannoprotein-based adhesion between bacteria and yeasts: whereas some research has shown the benefit of engineered adhesion to conjugation among bacteria (72), the native binding strategy employed in this work avoids the need to engineer recipient cells while allowing further options for engineering donors, e.g., by making the bacterial binding mechanism inducible. Additional work is also needed to discern the short-term dynamics of cell-cell adherence and the full extent to which mannose can reverse IDC-mediated perturbations. Previous research has demonstrated a wide repertoire of strategies *E. coli* use to derive resources from proximal producer cells, including through cell-cell nanotube connections (56, 73, 74). In our case, it remains unclear, however, whether $E_{cross}$ cells can acquire nutrients through conjugative pili, via diffusion from closer recipients or by killing clumped cells and importing lysed metabolites.

Despite still relatively low conjugation rates, we demonstrated that IDC can be applied to functionally alter recipient yeast populations. We "rescued" a low-growing population via IDC transfer of an essential gene, by depressing donor growth and making it dependent on recipient cells, in keeping with our dynamics findings. We also "killed" recipient cells via IDC-mediated Cas9 cutting of an essential gene. In this latter case, we deliberately added a layer of complexity unnecessary to the aim of killing cells, in that we designed yeast recipients to carry the essential gene URA3 episomally, so as to verify the cutting efficiency. Approaches aiming to depress recipient populations, without quantification of conjugation efficiency, could instead target an essential gene in the genome to induce cell death through the combination of double-strand break toxicity and removal of an essential gene.

Given the ubiquity of mannoproteins in fungal cells and the discrepancy in growth rates between prokaryotic and eukaryotic cells—which allow bacteria to adapt quickly and persist in adverse conditions—there is ample reason to believe that IDC could

be a viable strategy for altering additional fungal populations including pathogenic fungi such as *Candida glabrata*, *Malassezia restricta*, and *Aspergillus fumigatus*, for which treatments are limited and in great demand (2, 3, 75, 76). Our results in colonies suggest that IDC could be relevant in spatially constrained settings such as biofilms. Importantly, while colonies have been used as proxies for biofilm growth before (77, 78), they are extremely well controlled relative to natural biofilms which have greater diversity and lower nutrient availability. Considerable work is needed to determine the extent to which bacterial donors are able to intermix and conjugate in more realistic settings, such as porcine skin biofilms (79).

Perhaps, the most interesting avenue for future work would be targeting more complex genetic modification of recipient fungal cells, rather than simple killing or rescue. This could include more complex functions and a wider range of recipient species, such as modifying metabolic pathways in a consortium producing a useful product or disrupting quorum sensing function in virulent cells, for example, by targeting the farnesol pathway in *C. glabrata*. Importantly, while many of the findings here might be broadly relevant to other fungal species, our use of species-specific self-replicating plasmids presents a limitation. Additional fungal targets, such as undomesticated yeast and filamentous fungi, could benefit from recent work expanding possibilities for targeted chromosomal genetic insertions (80).

Recent work has demonstrated the capacity of conjugative DNA transfer to tune intercellular messages in a synthetic *E. coli* consortium (22), and such a strategy could feasibly be expanded to a wide range of both DNA programs and recipient species. This gets to the heart of IDC's power: unlike other perturbation strategies such as Type VI Secretion Systems, which have been used for targeted killing (81, 82), the possibilities for recipient programming via IDC are only as limited as our ability to engineer the DNA for those functions and express them in recipient populations, as well as the frequency of conjugative delivery. Our work here focuses primarily on the latter hurdle—achieving high-enough IDC rates to modify recipient populations and to control them tunably—but further work expanding the range of functions delivered to recipients could open the door to a vast array of *in situ* microbiome and synthetic consortia engineering.

## MATERIALS AND METHODS

### Strain and plasmid construction

Yeast cells in this study are derived from W303 strains developed by Müller et al. (*MAT***a** *can1-100 hmlaΔ::BLE leu9Δ::KANMX6 his3Δ::prACT1-ymCitrine-tADH::HIS3MX6*, with S288C version of *BUD1*) (51). Crossfeeding yeast strains ("S$_{cross}$," yMM1430) have additional mutations to make them auxotrophic for tryptophan and leucine overproducing: *LEU4$^{FBR}$ trp2Δ::NATMX4 URA3::prACT1yCerulean-tADH1*, with leucine feedback resistance (FBR) resultant from deletion of codon 548 of *LEU4*. S$_{cross}$ is also constitutively fluorescent for ymCitrine and yCerulean, whereas the baseline yeast used here (aka "WT yeast," "S," yMM1636) is only ymCitrine-fluorescent. Further mutations were introduced into these strains to make them auxotrophic for uracil and/or histidine, for IDC selection and CRISPR assay. Uracil was knocked out by amplifying a cassette of *URA3* homology arms, transforming into yMM1430, and selecting for growth on 5-fluoroorotic acid (5FOA). *HIS3* was replaced with either *KANMX6* or *HPHMX6*, depending on the strain (see Table S4 for the list of strains and related experiments), by amplifying either resistance gene with overlap for *HISMX6* regions.

Bacterial strains in this study are derived from Keio Collection strains of single-gene knockouts, based on BW25113 background [*F− Δ(araD-araB)567 lacZ4787Δ::*rrnB-3 λ− *rph-1 Δ(rhaD-rhaB)568 hsdR514*] (83). WT *E. coli* ("E") strains are simply BW25113 or Coli Genetic Stock Center (CGSC) #7636, containing different plasmids depending on the experiment (see Table S5 for list of plasmids and corresponding experiments). Crossfeeding mutations were introduced into CGSC #11110 (*ΔtrpR789:kan$^R$*), which lacks the trp repressor gene and has been shown to be tryptophan overproducing (56). Briefly, the

kanamycin resistance gene at the *trpR* locus was "flipped" out via flippase recognition target sequences and flippase-expressing plasmid pMM0821 (84). Leucine auxotrophy was introduced by λred recombination of PCR-amplified Δ*leuA781::kan^R*, from CGSC #8373, using pMM0820, which expresses genes for λred. *kan^R* was again flipped out to obtain kMM127, a double knockout of Δ*trpR*, Δ*leuA*, with no antibiotic resistance. Note that we originally constructed the crossfeeding *E. coli* ("E_cross") from Δ*leuA::kan^R* (CGSC #8373), but it caused severe aggregation in coculture, such that cells would precipitate out of media immediately, whereas the same mutation introduced from the Δ*trpR::kan^R* strain did not produce this result. Moreover, we found that Δ*leuB::kan^R* (CGSC #11943) proved prototrophic for leucine over long time periods, despite its similar function in the leucine biosynthesis pathway (ref).

IncP-type IDC plasmids (85) pTA-Mob 1.0 (*trans*-transferring) and pTA-Mob 2.0 (*cis*-transferring) were generously provided to us by the Karas lab (58). pTA-Mob 2.0 contains gentamicin resistance for bacterial selection, *URA3* and *HIS3* genes for yeast selection, *CEN6/ARSH4* for yeast maintenance, and the *ori^T* sequence required for conjugative transfer of the plasmid into recipients, whereas pTA-Mob 1.0 only carries gentamicin resistance. Constitutive bacterial reporter pMM0819 contains *pProD:mCherry*, using a synthetic reporter meant to be highly expressing and minimally susceptible to the cell phase (86, 87). IDC plasmids for *trans*-transfer were constructed using the Golden Gate-based Yeast MoClo Toolkit (88) (YTK), to modularly assemble a fluorescent yeast reporter (*pTDH3-yeBFP*), IDC selection (*HIS3*), and yeast replication machinery (*CEN6/ARSH4*). The *ori^T* sequence was then added to the connector sequence downstream of yeBFP via Gibson assembly.

For IDC-mediated CRISPR killing assay, the S_cross, *ura3Δ0 hismx6Δ::HPHMX6* strain yMM1786 was transformed with a plasmid containing *pTDH3-yeBFP URA3 CEN/ARS*. sgRNAs were designed to cut within the connector region of this plasmid (ConR1 from YTK), downstream of yeBFP, such that any YTK-assembled plasmid containing the ConR1 sequence could be a target in future experiments. CRISPR plasmids were assembled using Ellis lab plasmids (89, 90). Briefly, oligos for five sgRNA sequences targeting the ConR1 region were designed using Benchling (91), PNK phosphorylated, and annealed. Annealed oligos were then assembled into sgRNA entry vector pMM1340 via Golden Gate assembly and transformed into bacteria, selecting with carbenicillin. Purified and sequence-verified sgRNA plasmids were then digested with EcoRV to isolate the sgRNA sequences with homology arms matching the insertion site of the Cas9 plasmid. Plasmid pMM1341, which contains Cas9, GFP, and *HIS3*, was digested with BsmBI to remove GFP and leave homology arms for sgRNA at each end of the resultant linear DNA. The two pieces were combined via yeast recombinant cloning. Finally, *ori^T* was inserted by ligating a modified version of the *ori^T* sequence with the assembled Cas9-sgRNA after digesting with AatII and SacII.

## Batch culture experiments and IDC counting

Yeast and bacterial cultures used in each batch culture experiment were grown overnight in selective YPD or LB media, at 30°C or 37°C, respectively. After ≥16 hours' growth, bacterial strains were measured for optical density at λ = 600 nm (OD600), yeast strains were measured for OD660, and each culture was washed at least two times with SC or M9 sans glucose or amino acids. Cells were then combined such that each reaction started with 1E7 cells, based on OD measurements. Growth media were composed of 200 µL of a 75:25 mixture of SC:M9 minimal media (see supplemental discussion) with 2% glucose, appropriate amino acids, and antibiotics to maintain each bacterial plasmid. Amino acid percentages in the text are based on the following molarities, considered 100%: L = 762 µM, W = 245 µM, U = 178 µM, and H = 95.4 µM. For clumping experiment (Fig. 3), half of the media was supplemented with 4% mannose. Upon spiking cells into 96-well CellVis back-walled optical glass-bottom plates (cat #P96-1-N), plates were sealed with gas permeable membrane (Fisher Scientific cat #50-550-304) seals to allow air flow for aerobic conditions.

Plates were grown in a customized Tecan Fluent automated plate handling robot, on a Bioshakes heater-shaker, kept at 30°C and rotating at 1,000 rpm with a 2-mm orbital. In 15-minute intervals, the Fluent was programmed to transfer each 96-well plate to a connected Tecan Spark fluorimeter, in which each well was measured for OD600, mCherry (Ex = 575 nm, Em = 620 nm, 20 nm bandwidth, and gain = 60), ymCitrine (Ex = 500 nm, Em = 545 nm, 20 nm bandwidth, and gain = 60), and, for CRISPR experiment (Fig. 7), yeBFP (Ex = 381 nm, Em = 445 nm, 20 nm bandwidth, and gain = 60). After each plate was measured, it was returned to the Bioshakes, where it grew for another 15 minutes until the next read. Each plate was grown in this way for roughly 18–24 hours, at which time plates were briefly spun (1 min at $1,000 \times g$) to remove droplets from plate seal. Each plate was then diluted 1:10 in fresh media (180 µL media + 20 µL previous day's culture) for that day's growth, with another 20 µL diluted into a plate of PBS + 0.1% Tween for flow cytometry (see below). Tecan data were consolidated in Excel format and imported into MATLAB via a custom script, which parses the Tecan Excel export format based on the number of plates and channels measured. All further analyses were performed in MATLAB, including normalization, in which all fluorescence measurements were divided by the max reading of that channel; these normalized reads were used for D:R ratios in Fig. 2; Fig. S3.

An additional 100 µL of each day's culture was added, undiluted, to a 24-well plate containing IDC-selective SC with 2% agar: for *cis*-transfer experiments (Fig. 2, 5 and 6), SC-UH was used, whereas SC-H was used for *trans*-transfer experiments (Fig. 3, 5 and 7). IDC plates were then placed in a culture shaker at 30°C for ~40 minutes, without lids, to dry. Once dried, IDC plates were incubated for ~3 days to grow countable transconjugant colonies. Individual transconjugant colonies were counted for CFU, unless wells were saturated, for which estimates were generated based on the density relative to countable wells, up to 500, the value assigned to lawns; while these represent a minority of measurements, counts ≥ 200 CFU in Fig. 2 and 5 should be considered estimates and likely undercounts. Samples with no transconjugants were set to 0.1 in log-scale plots, to account for detection limit. For rescue assay, due to higher counts, cultures after day 2 were serially diluted up to 1:10,000, in increments of 10× dilutions, and frogged onto SC-UH 2% agar in a 245-mm BioAssay Dish (Corning cat # 431111). Countable microcolonies from frogging dilutions were averaged, based on dilution value; thus, saturated microcolonies were ignored.

## Colony experiments

Each strain was grown, measured, washed, and diluted as in batch culture experiments. Because we wanted 2 µL mixed culture droplets to seed each colony, we had to lower the input cell counts to 1E6 of each cell type. Strains were combined accordingly and then aliquoted into strip tubes, from which we were able to multichannel pipette ≥ 18 identical 2 µL mixed colonies onto 2% agar minimal media plates. Each 60-mm plate (Eppendorf cat #0030701011) contained 75:25 SC:M9 with appropriate bacterial antibiotics for plasmid maintenance, 2% agar, and one concentration of amino acids, such that each plate represented a single experimental condition; molten media were aliquoted to plates in equal (15 mL) portions. Once mixed colonies were added to plates, they were allowed to grow at 30°C for 6 days. Three representative colonies (by eye) were designated after the first day's growth to be repeatedly imaged over the entire time course, while another three were designated to be imaged that day only, after which they would be scraped, washed, and measured by flow cytometry and IDC plating. All colonies were numbered, upon being selected, to correlate measurements.

Plates were imaged for fluorescence using a Zeiss AxioZoom V16 dissecting microscope, at UW-Madison's Newcomb Imaging Center. Each *cis*-donor mixed colony was imaged for mCherry (Zeiss Set 43 BP 545/25, FT 570, BP 605/70, 200 ms exposure) and ymCitrine (Zeiss Set 46 HE, EX BP 500/20, BS FT 515, EM BP 535/30, 600 ms exposure), while *trans*-donor mixed colonies were additionally measured for yeBFP (Zeiss set 49:

G365, FT395, BP445/50, 500 ms exposure). All images were taken at 8× zoom. See supplemental discussion for more information on image processing and analysis.

After imaging, colonies were manually scraped off plates via micropipette tips and diluted into 1.5-mL tubes containing 1 mL water. Each diluted colony was vortexed for ~30 seconds to break up colonies and dilute residual agar and then spun at 3,000 × $g$ for 5 min. Eight hundred microliters of water was removed from each tube, and cells were resuspended in the remaining ~200 µL, 100 µL of which was plated for IDC selection (see "Batch culture experiments and IDC counting) and another 20 µL was aliquoted into 180 µL PBS + 0.1% Tween for flow cytometry (see below). After 6 days of growth, the colonies (1–3) designated for continual microscopy imaging were processed and measured in this way.

## Flow cytometry

After diluting cells from culture or colonies (see above) 1:10 into PBS + 0.1% Tween (total volume = 200 µL) in 96-well round-bottom plates (Fisher Scientific cat #07-200-760), samples were measured for cellular composition using a Thermo Fisher Attune NxT V6 Flow Cytometer at UW-Madison's Carbone Cancer Center, which includes a 96-well compatible autosampler. Because the sizes of bacteria and yeast are so different, each coculture was measured twice, with different forward and side scatter voltages for each cell type (monoculture controls were generally measured using only that species' voltage settings, though at least two of the other species were included for each to get baseline counts). Each well was measured for ymCitrine (488 nm laser, 530/30 503LP filters, off target fluorescent) and mCherry (561 nm laser, 620/15 600LP filters), in addition to scatter, using a draw volume of 20 µL, at a flow rate of 200 µL/min.

FCS files exported from the Attune were processed via custom MATLAB tools modified for dual-voltage experiments. Gates were drawn per voltage setting to capture all cells of that species based on fluorescence and forward scatter. FCS files were imported, correlated with sample information, and queried for inclusion in each gate. Summary tables for each cell type were consolidated to combine all readings per experiment, upon which noise floors were calculated based on negative controls per voltage setting. Gate-defined cell counts for each species were subtracted by these baselines and converted to total cells per 100 µL, to compare to IDC counts (see IDC prep in "Batch culture experiments and IDC counting"). Cell counts were further normalized by dividing by the max count for that experiment and cell type; normalized counts were used to generate D:R ratios (Fig. 2, 5, and 6).

## Microscopy of culture aggregates

Batch culture samples were diluted to various degrees (depending on day and sample density) in media lacking glucose and amino acids, but with mannose for samples grown with it, in a CellVis 96-well back-walled optical glass-bottom plates (cat #P96-1-N). Plates were loaded onto the stage of an inverted fluorescence microscope (Nikon TiE), enclosed by an opaque incubation chamber. A custom Nikon JOBS script was written to image each well of a plate in three random locations distal to the well edges, with a 2-second wait time before each photo to allow cells to settle after moving the stage. All wells were imaged at 10× objective for mCherry (Chroma 96365, Ex = 560/40×, Em = 630/75 m, and 200 ms exposure), ymCitrine (Chroma 96363, Ex = 500/20×, Em = 535/30 m, and 600 ms exposure), and yeBFP (Chroma NC296093, Ex = 350/50×, Em = 460/50 m, and 500 ms exposure). See supplemental discussion for image processing and analysis.

## Statistics

Two-sample $t$-tests calculated using MATLAB's "ttest2" function, with default two tails (Fig. 3c, 5c and 7c). One-way ANOVA of ICQ calculated between coculture pairings in Fig. 5c via MATLAB's "anovan" function, using type II sum of squares, alpha = 0.5 (significance level ≤ 0.5). See Table S6 for $F$-values and degrees of freedom for one-way ANOVAs for

Fig. 5c. Significance stars indicate *P*-values less than 0.05 (*), 0.01 (**), or 0.005 (***) or else are considered not significant. The number of replicates is cited in each figure caption.

## Data analysis and figures

Unless otherwise specified, all data processing was performed using custom MAT-LAB scripts, which can be accessed at the GitHub repository for this paper: https://github.com/mccleanlab/Stindt_2023. Most data plots were generated with the gramm MATLAB toolbox (92), and flow diagrams were created with BioRender.com.

## ACKNOWLEDGMENTS

We would like to thank Taylor Scott and Kieran Sweeney for the invaluable feedback on this work and the help with coding and the McClean Lab for many helpful discussions. Conjugative plasmids pTA-Mob were generously provided to us by the Karas Lab, Western University, Ontario, Canada; many thanks are due to Bogumil, Ryan Cochrane, and Maximillian Soltysiak for their generosity in mailing and emailing.

This work was supported by NIGMS (MIRA R35GM128873), NIAID (R01AI154940), and the Wisconsin Alumni Research Foundation (WARF 135 AAI9593). Flow cytometry was enabled by the University of Wisconsin Carbone Cancer Center Support Grant (P30CA014520). Megan Nicole McClean, PhD, holds a Career Award at the Scientific Interface from the Burroughs Wellcome Fund. Kevin R. Stindt was supported in part by the Molecular Biophysics Training Grant (T32GM130550).

K.R.S. and M.N.M. planned the experiments. K.R.S. performed the experiments. K.R.S. and M.N.M. analyzed the data and wrote the paper.

## AUTHOR AFFILIATIONS

[1]Department of Biomedical Engineering, University of Wisconsin-Madison, Madison, Wisconsin, USA
[2]Doctoral Program in Biophysics, University of Wisconsin-Madison, Madison, Wisconsin, USA
[3]University of Wisconsin Carbone Cancer Center, University of Wisconsin School of Medicine and Public Health, Madison, Wisconsin, USA

## AUTHOR ORCIDs

Kevin R. Stindt  http://orcid.org/0000-0002-6028-3773
Megan N. McClean  http://orcid.org/0000-0002-1141-4802

## FUNDING

| Funder | Grant(s) | Author(s) |
|---|---|---|
| HHS \| NIH \| National Institute of General Medical Sciences (NIGMS) | R35GM128873 | Kevin R. Stindt |
| | | Megan N. McClean |
| HHS \| NIH \| National Institute of Allergy and Infectious Diseases (NIAID) | R01AI154940 | Kevin R. Stindt |
| | | Megan N. McClean |
| Wisconsin Alumni Research Foundation (WARF) | 135AAI9593 | Kevin R. Stindt |
| | | Megan N. McClean |
| University of Wisconsin Carbone Cancer Center (UWCCC) | P30CA014520 | Kevin R. Stindt |
| | | Megan N. McClean |
| Molecular Biophysics Training Grant | T32GM130550 | Kevin R. Stindt |

## DATA AVAILABILITY

The data that support the findings of this study are openly available in Dryad at https://doi.org/10.5061/dryad.5mkkwh7c7.

## ADDITIONAL FILES

The following material is available online.

### Supplemental Material

**Supplemental Information (mSystems00050-24-s0001.pdf).** Extended methods and supplemental figures.

### Open Peer Review

**PEER REVIEW HISTORY (review-history.pdf).** An accounting of the reviewer comments and feedback.

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
