## [Reviewer comments · mSystems]

Tuning interdomain conjugation to enable *in situ* population modification in yeast

Kevin Stindt and Megan McClean

Corresponding Author(s): Megan McClean, University of Wisconsin-Madison

Review Timeline:

Submission Date:	January 9, 2024
Editorial Decision:	March 3, 2024
Revision Received:	April 12, 2024
Accepted:	April 15, 2024

Editor: Babak Momeni

Reviewer(s): The reviewers have opted to remain anonymous.

Transaction Report:

DOI: <https://doi.org/10.1128/msystems.00050-24>

Re: mSystems00050-24 (Tuning interdomain conjugation to enable *in situ* population modification in fungi)

Dear Prof. Megan Nicole McClean:

Revision Guidelines

Sincerely,
Babak Momeni
Editor
mSystems

Reviewer #1 (Comments for the Author):

This study by Stindt and McClean is a tour-de-force. It demonstrates an extremely comprehensive quantitative analysis of E. coli-to-yeast interdomain conjugation (IDC) in response to different environmental conditions. It also shows different ways by which IDC can be used to control population dynamics.

In the past several years, there have been increasing efforts and measure gene transfer mediated by conjugation (e.g., Lopatkin

et al, Nat Microbio, 2016) and demonstrating how gene transfer can be useful in controlling the function of microbial communities (Wang et al, Nat Chem Biol 2022). However, most of these studies have focused on gene transfer between populations of the same species. That being said, I think the authors might want to cite these relevant studies.

While IDC has been established previously, detailed quantifications of its dynamics are lacking. Yet, such measurements are critical for understanding the role of IDC in nature and for using IDC for engineering applications. This study fills that gap. In addition, the work demonstrates how IDC can be used to tune the dynamics of pair-wise microbial communities. On the technical side, the work is incredibly comprehensive. I anticipate that both the experimental system and the generated data will be valuable for the broad research community.

I am enthusiastically supportive of the publication of the work, with very minor revisions on presentation. Given the extremely comprehensive nature of the study, I don't have any suggestions for (or the need for) additional experiments or modeling analysis.

Some very minor points.

Many of the figures, especially those generated from modeling analysis, are hard to read. One example is Figure 6C. While I understand the message, I wish the data are more clearly presented by improving the visual.

I wish to see a clearer description of how parameter fitting was carried out, as well as a more clear description on the quality of fit. If I understood correctly, they generated extensive sampling of the parameter space (within reasonable ranges), did forward simulations, and found the parameter set that generated the closest match with experimental data. This parameter set was then used as the estimated set.

Reviewer #2 (Comments for the Author):

This paper explores how transfer of DNA from bacterial cells to yeast cells can be used to manipulate bacteria-yeast interactions and population dynamics. Much of the work is proof of principle in terms of showing under what conditions these cross-species transfers might work and what can be done to the interactions and population dynamics using interdomain conjugation (IDC). I appreciate how the authors did a range of experiments to understand the conditions that change the rate of IDC based on manipulating the *E. coli*-*S. cerevisiae* interactions. Once they did this initial experimental work, they then turned to deterministic models to more fully explore the "tuning space" of this system to optimize IDC outcomes. I am less familiar with this modeling approach and will focus my review on the experimental approaches. The authors also explored how spatial structure (growing in biofilm spots as opposed to well-mixed liquid cultures) affected IDC outcomes. With all of this optimization, they then used the best conditions to use IDC to affect (both positively and negatively) yeast population dynamics. I appreciated the full integration of all of these approaches and this integration makes the paper a great fit for mSystems.

As someone who is somewhat outside of the center of this research area, I really appreciated how clearly the paper was written and the thoughtfulness put into the figures. For some experts in the field, some of the introduction text may be a bit too detailed (e.g. how conjugative transfer of DNA works). But mSystems has a wide readership and this work will be of interest to a broad audience. So I would recommend that the authors not remove any of that very helpful context in the introduction.

One somewhat minor issue I had with overall framing and language used in the paper is that the work implies that this interdomain transfer could occur between bacteria and all fungi. But I think the reality is that this type of interaction could only really occur between bacteria and yeast. It is hard to imagine the same process happening between bacteria and filamentous fungi because of their very different ways of growing, issues with getting plasmids into filamentous fungi, etc. I suggest that the authors consider changing some instances of "fungi" (title, abstract, and some other places) to yeast unless they have strong arguments against that.

I appreciate that the authors hint at potential ways this might be used to manipulate fungal pathogens such as *Candida* species in their Discussion. But I wonder if this is premature or should have a lot more caveats. The colony biofilms they were using in their experiments were relatively "happy" cells at very high densities without any other microbes present. None of this would be true in more realistic conditions of a human microbiome where cell densities are much lower, cells are in very different growth states, and many other microbes may be present. The authors should either tone down the alluding to potential applications or add in some notes about these challenges/caveats. Additionally in this section, the authors might want to consider applications of this approach to systems where their lab conditions might be more relevant: in industrial or food production settings where fungi and bacteria can reach high concentrations and are often nearly pure cultures.

In lines 178-183 - I was a bit disappointed that the authors did not do a few more experiments to figure out how the *E. coli* and *S. cerevisiae* were not able to support each other's growth when both leucine and tryptophan were not present. They suggest this may be due to competition for glucose, but understanding this lack of coexistence of the two microbes might actually help the authors have a wider set of conditions in which to do their "tuning." Experiments where they manipulated glucose or tried to identify some other antagonistic mechanism of interaction would have been helpful here.

Line 549: "Individual transconjugant colonies were counted for CFU, unless wells were saturated..." I am a bit confused on the "for which estimates were generated based on density relative to countable wells" Can you explain the rationale for this approach a bit more? How often did this happen? How might this approach affect your estimates of transconjugants?

Figure 5B legend. Change "yeasts" to "*S. cerevisiae*" and "bacteria" to *E. coli* to be more specific. Do this elsewhere in other figure legends as well.

Figure 4C: Is the x-axis cut off?

Minor thing: I found it a bit clunky to follow the figures since they were all split up into separate elements of one main figure. If you can combine them into a single multi-panel figure, that would be helpful.

Line 524: Why 4% mannose? Did you try a range of mannose concentrations or is this somehow known to be the best concentration to use? Provide some rationale/justification in the methods.

Reviewer #1 (Comments for the Author):

This study by Stindt and McClean is a tour-de-force. It demonstrates an extremely comprehensive quantitative analysis of *E. coli*-to-yeast interdomain conjugation (IDC) in response to different environmental conditions. It also shows different ways by which IDC can be used to control population dynamics.

In the past several years, there have been increasing efforts to measure gene transfer mediated by conjugation (e.g., Lopatkin et al, Nat Microbio, 2016) and demonstrating how gene transfer can be useful in controlling the function of microbial communities (Wang et al, Nat Chem Biol 2022). However, most of these studies have focused on gene transfer between populations of the same species. That being said, I think the authors might want to cite these relevant studies.

While IDC has been established previously, detailed quantifications of its dynamics are lacking. Yet, such measurements are critical for understanding the role of IDC in nature and for using IDC for engineering applications. This study fills that gap. In addition, the work demonstrates how IDC can be used to tune the dynamics of pair-wise microbial communities. On the technical side, the work is incredibly comprehensive. I anticipate that both the experimental system and the generated data will be valuable for the broad research community.

I am enthusiastically supportive of the publication of the work, with very minor revisions on presentation. Given the extremely comprehensive nature of the study, I don't have any suggestions for (or the need for) additional experiments or modeling analysis.

We appreciate the reviewer's excellent summary of our work. We have improved our introduction to the field by evaluating and including additional relevant literature, including the suggested references. In addition, we have revised figures and text to better illustrate and describe the modeling efforts.

Some very minor points.

Many of the figures, especially those generated from modeling analysis, are hard to read. One example is Figure 6C. While I understand the message, I wish the data are more clearly presented by improving the visual.

The reviewer's comments regarding clarity of the model analysis (i.e. Figure 6, fitting procedures) are well taken. To clarify Figure 6D (formerly Figure 6C), we have revised the figure to include only the most illustrative pairs of yeast and bacteria. (All pairs are available in a new Supplemental Figure SF15). We now indicate corresponding experiment conditions using cyan boxes. To connect to the phase diagram in Figure 6E, we highlight the optimal IDC zone in Figure 6D using dashed lines. We revised the axes to be less busy and include additional labels.

I wish to see a clearer description of how parameter fitting was carried out, as well as a more clear description on the quality of fit. If I understood correctly, they generated extensive sampling of the parameter space (within reasonable ranges), did forward simulations, and found

the parameter set that generated the closest match with experimental data. This parameter set was then used as the estimated set.

We thank the reviewer for the suggestion, and the reviewer is correct in their summary of model fitting and ranking. We've included language in the Supplementary Information that better describes the calculation of model error (RSS) and have included these RSS values in the model fit figures, SF8 and SF10. The RSS values provide a relative sense of which measurements were better predicted.

Reviewer #2 (Comments for the Author):

This paper explores how transfer of DNA from bacterial cells to yeast cells can be used to manipulate bacteria-yeast interactions and population dynamics. Much of the work is proof of principle in terms of showing under what conditions these cross-species transfers might work and what can be done to the interactions and population dynamics using interdomain conjugation (IDC). I appreciate how the authors did a range of experiments to understand the conditions that change the rate of IDC based on manipulating the E.coli-S. cerevisiae interactions. Once they did this initial experimental work, they then turned to deterministic models to more fully explore the "tuning space" of this system to optimize IDC outcomes. I am less familiar with this modeling approach and will focus my review on the experimental approaches. The authors also explored how spatial structure (growing in biofilm spots as opposed to well-mixed liquid cultures) affected IDC outcomes. With all of this optimization, they then used the best conditions to use IDC to affect (both positively and negatively) yeast population dynamics. I appreciated the full integration of all of these approaches and this integration makes the paper a great fit for mSystems.

As someone who is somewhat outside of the center of this research area, I really appreciated how clearly the paper was written and the thoughtfulness put into the figures. For some experts in the field, some of the introduction text may be a bit too detailed (e.g. how conjugative transfer of DNA works). But mSystems has a wide readership and this work will be of interest to a broad audience. So I would recommend that the authors not remove any of that very helpful context in the introduction.

We appreciate the reviewer's comments, generous summary, and the opportunity to clarify the paper. We have addressed the reviewer's specific feedback below.

One somewhat minor issue I had with overall framing and language used in the paper is that the work implies that this interdomain transfer could occur between bacteria and all fungi. But I think the reality is that this type of interaction could only really occur between bacteria and yeast. It is hard to imagine the same process happening between bacteria and filamentous fungi because of their very different ways of growing, issues with getting plasmids into filamentous fungi, etc. I suggest that the authors consider changing some instances of "fungi" (title, abstract, and some other places) to yeast unless they have strong arguments against that.

We have addressed this point by changing the title and text from "...fungi" to "...yeasts" as appropriate. We changed "fungi" to "yeast" particularly when using "fungi" incorrectly implied that the results and scope of this paper apply to all fungi. We left "fungi" when it was used in the context of broadly discussing future research on and applications for interdomain conjugation. Indeed, in our first submission we went back and forth between the terms, as we expect many of these results to be applicable to other fungi. However, as the reviewer mentions, filamentous fungi (at least) pose specific challenges. We have expanded the discussion to be more specific about how broadly these results might be extrapolated and how we might address application to non-yeast fungi, including filamentous fungi.

I appreciate that the authors hint at potential ways this might be used to manipulate fungal pathogens such as *Candida* species in their Discussion. But I wonder if this is premature or should have a lot more caveats. The colony biofilms they were using in their experiments were relatively "happy" cells at very high densities without any other microbes present. None of this would be true in more realistic conditions of a human microbiome where cell densities are much lower, cells are in very different growth states, and many other microbes may be present. The authors should either tone down the alluding to potential applications or add in some notes about these challenges/caveats. Additionally in this section, the authors might want to consider applications of this approach to systems where their lab conditions might be more relevant: in industrial or food production settings where fungi and bacteria can reach high concentrations and are often nearly pure cultures.

*These points are well taken, and we have adjusted our discussion accordingly. We expanded discussion of the caveats to the colony work, specifically listing differences between colonies and natural biofilms. While colonies have been used as proxies for biofilms (for which we added references), we further discuss how this research might progress towards more realistic biofilm models, including porcine skin models of *Candida* infections. We concur with the reviewer that more immediate applications of this work would be in bioproduction settings and have included these applications in the Introduction to the paper.*

In lines 178-183 - I was a bit disappointed that the authors did not do a few more experiments to figure out how the *E. coli* and *S. cerevisiae* were not able to support each other's growth when both leucine and tryptophan were not present. They suggest this may be due to competition for glucose, but understanding this lack of coexistence of the two microbes might actually help the authors have a wider set of conditions in which to do their "tuning." Experiments where they manipulated glucose or tried to identify some other antagonistic mechanism of interaction would have been helpful here.

The reviewer makes an excellent point here. We were similarly disappointed that obligate mutualism proved unsustainable. There are many possible factors contributing to antagonism between the strains (e.g. glucose competition, pH). Therefore, because the main purpose of the crossfeeder strains was to allow population tuning, we left investigation of stable coexistence to future work. We completely agree that such work would be valuable. To highlight these important considerations, we have expanded on the possible reasons for antagonism between the strains, particularly as it relates to obligate mutualism, in the Supplemental information section on condition testing.

Line 549: "Individual transconjugant colonies were counted for CFU, unless wells were saturated..." I am a bit confused on the "for which estimates were generated based on density relative to countable wells" Can you explain the rationale for this approach a bit more? How often did this happen? How might this approach affect your estimates of transconjugants?

This estimation scheme was the case for a minority of IDC measurements and was used to record IDC counts that couldn't easily be discerned as individual CFUs on a 24-well plate of agar media. For every experiment that had a handful of high IDC counts we counted colonies up to at least 200 per well and used those counts to approximate the coverage of wells by denser colony growth. The value 500 was used to represent lawns. We only used these estimates for conditions where the underlying conclusions were not changed by greater resolution in the upper IDC range and where the estimation provides an underestimate of true IDC. For experiments in which many samples gave high IDC (e.g. rescue) we increased measurement sensitivity to capture nuances at the high end of IDC (e.g. by frogging dilutions).

As an example, here is some unpublished data of both measured and estimated IDC CFUs from a preliminary experiment. Wells in which CFUs were counted are highlighted in blue, while wells that were estimated are highlighted in orange. Note too that a handful of samples are resolved (counted) > 200 CFU, and these were used to inform estimates at and above this range.

Fig 5d is a good example of use of these estimates in the paper. The IDC counts above 200 are quantized to the nearest hundred value (300, 400, 500). We specified the extent of this estimation with references to Figures in the methods as well as in Figure legends. These estimations are undercounts of true IDC values, based on comparisons between these estimates and serially-diluted IDC measurements for which we obtained granular counts. Underestimating the IDC means that the plot in Figure 5d could be slightly steeper, actually strengthening our conclusions.

Figure 5B legend. Change "yeasts" to "S. cerevisiae" and "bacteria" to E. coli to be more specific. Do this elsewhere in other figure legends as well.

We thank the reviewer for pointing this out, and have made the requested changes to all Figures and SI Figs.

Figure 4C: Is the x-axis cut off?

Yes, due to a formatting error the x-axis was cut off in the submitted version of the paper. We have repaired the figure in the revised submission.

Minor thing: I found it a bit clunky to follow the figures since they were all split up into separate elements of one main figure. If you can combine them into a single multi-panel figure, that would be helpful.

We have submitted figures as single multi-panel figures in the revised submission.

Line 524: Why 4% mannose? Did you try a range of mannose concentrations or is this somehow known to be the best concentration to use? Provide some rationale/justification in the methods.

Our use of 4% mannose was based on the work of Scarinci and Sourjik (ISME 2023), in which they used this percentage to disrupt mannoprotein adhesion between the species. In this paper, we used this concentration to produce a strong effect to test “on” vs “off” for adhesion. Future work will include comparisons of adhesion strength and frequency (via mannose dosage) and timing factors (e.g. timing of mannose addition) that modulate IDC.

Re: mSystems00050-24R1 (Tuning interdomain conjugation to enable *in situ* population modification in yeast)

Dear Prof. Megan Nicole McClean:

Your manuscript has been accepted, and I am forwarding it to the ASM production staff for publication. Your paper will first be checked to make sure all elements meet the technical requirements. ASM staff will contact you if anything needs to be revised before copyediting and production can begin. Otherwise, you will be notified when your proofs are ready to be viewed.

Cover Image Submissions: If you would like to submit a potential Cover Image, please email a file and a short legend to msystems@asmusa.org. Please note that we can only consider images that (i) the authors created or own and (ii) have not been previously published. By submitting, you agree that the image can be used under the same terms as the published article. Image File requirements: TIF/EPS, 7.5 inches wide by 8.25 inches tall (at least 2,250 pixels wide by 2,475 pixels tall), minimum 300 dpi resolution (600 dpi preferred), RGB, and no figure elements, e.g., arrows or panel labels. The legend should be a short description of the image, 1-2 sentences recommended.

Sincerely,
Babak Momeni
Editor